# Metabolism of Reactive Oxygen Species in Osteosarcoma and Potential Treatment Applications

**DOI:** 10.3390/cells9010087

**Published:** 2019-12-30

**Authors:** Wei Sun, Bing Wang, Xing-Long Qu, Bi-Qiang Zheng, Wen-Ding Huang, Zheng-Wang Sun, Chun-Meng Wang, Yong Chen

**Affiliations:** 1Department of Musculoskeletal Oncology, Fudan University Shanghai Cancer Centre, Department of Oncology, Fudan University Shanghai Medical School, Shanghai 200032, China; wsun14@fudan.edu.cn (W.S.); Zhengbqk@hotmail.com (B.-Q.Z.); orienthwd@163.com (W.-D.H.); specialsamsun@126.com (Z.-W.S.); cmwang1975@163.com (C.-M.W.); 2Department of Oncological Surgery, Minhang Branch, Shanghai Cancer Center, Fudan University, Shanghai 200240, China; bingwang1101@163.com (B.W.); quxl681@163.com (X.-L.Q.)

**Keywords:** osteosarcoma, ROS, HIF-1α, FoxO1, microarray

## Abstract

Background: The present study was designed to explore the underlying role of hypoxia-inducible factor 1α (HIF-1α) in reactive oxygen species (ROS) formation and apoptosis in osteosarcoma (OS) cells induced by hypoxia. Methods: In OS cells, ROS accumulated and apoptosis increased within 24 h after exposure to low HIF-1α expression levels. A co-expression analysis showed that HIF was positively correlated with Forkhead box class O1 (FoxO1) expression and negatively correlated with CYP-related genes from the National Center for Biotechnology Information’s Gene Expression Omnibus (NCBI GEO) datasets. Hypoxia also considerably increased HIF-1α and FoxO1 expression. Moreover, the promoter region of FoxO1 was directly regulated by HIF-1α. We inhibited HIF-1α via siRNA and found that the ROS accumulation and apoptosis induced by hypoxia in OS cells decreased. In this study, a murine xenograft model of BALB-c nude mice was adopted to test tumour growth and measure the efficacy of 2-ME + As_2_O_3_ treatment. Results: Ad interim knockdown of HIF-1α also inhibited manganese-dependent superoxide dismutase (MnSOD), catalase and sestrin 3 (Sesn3) expression in OS cells. Furthermore, hypoxia-induced ROS formation and apoptosis in OS cells were associated with CYP450 protein interference and were ablated by HIF-1α silencing via siRNA. Conclusions: Our data reveal that HIF-1α inhibits ROS accumulation by directly regulating FoxO1 in OS cells, which induces MnSOD, catalase and Sesn3 interference, thus resulting in anti-oxidation effects. The combination of an HIF-1α inhibitor (2-mercaptoethanol,2-ME) and ROS inducer (arsenous oxide, As_2_O_3_) can prohibit proliferation and migration and promote apoptosis in MG63 cells in vitro while inhibiting tumour growth in vivo.

## 1. Introduction

Osteosarcoma (OS) is the most common primary bone cancer and is one of the leading causes of mortalities related to cancer in paediatric patients [1]. The five-year survival rate has increased significantly due to recent progress in current therapies, particularly the introduction of combining surgical incision and adjuvant chemotherapy and radiotherapy [2]. Nevertheless, the prognosis of patients with recurrent disease or lung metastasis remains poor, and due to the existence of confounding benign situations, such as growing pains in puberty, the general dictation rate for osteosarcoma remains low [3]. Thus, it is essential to further explore the underlying molecular mechanism of OS and develop novel therapeutic targets. 

The subgroup of Forkhead box class O (FoxO) transcription factors contains important regulators of the genome; they are characterized by the structural feature of a winged-helix in their DNA binding domain [4,5]. The mammalian gene of the proteins encodes four different variants (FoxO1, Fox3a, FoxO4 and FoxO6). Epigenetic regulation, especially phosphorylation by AKT serine/threonine kinase (Akt), could promote its degradation and dysfunction [6]. Importantly, FoxO factors have been identified as tumour suppressors involved in the biological behaviours of cancer cells, such as extracellular matrix degradation, angiogenesis and migration [7,8,9]. FoxO factors are involved in regulating the genes associated with cellular cycle progression and inducing apoptosis [10]. It has also been proven that FoxO proteins play a suppressive role in cancers of multiple systems, such as gastric cancer, bladder cancer, liver cancer and breast cancer [8,11,12,13]. FoxO factors are also involved in regulating reactive oxygen species (ROS) detoxification by upregulating mitochondrial superoxide dismutase (SOD2) [14].

Mitochondria are centres for cellular bioenergetic activities and are important sources for ROS [15]. ROS production occurs via respiratory complexes located in the inner mitochondrial membrane. Mitochondrial ROS can contribute oxidative stress to mitochondrial DNA (mtDNA), proteins and lipids, but they are also involved in many signalling pathways from the mitochondria to the cytoplasm. However, the role of mitochondrial ROS is varied within different types of cancer; in some cancers such as lung cancer, upregulation could lead to increased cell proliferation [16,17]. The functional regulation of mitochondria is vital to normal and tumour cells [18].

The genome within the mitochondria encodes 12 proteins closely associated with biogenetic activities [19]. The expression of these genes is regulated by a network of transcription factors, such as *Saccharomyces cerevisiae* and mitochondrial transcription specificity factors, in up- and downstream signalling pathways. All of the above transcription factors bind to and are further regulated by nuclear respiratory factors and PGC-1 family coactivators [20,21]. In addition, the transcription factor c-Myc could participate in the stress response to dysfunctional mitochondria [20]. Interestingly, hypoxia-inducible factor 1α (HIF-1α) could also contribute to the downregulation of mitochondrial biogenesis by inhibiting c-Myc as part of the cellular response to environment alterations.

We have previously reported that FoxO1 could promote the expression of antioxidant proteins such as MnSOD, catalase and Sesn3 [22]. Specifically, FoxO1 expression is driven by HIF-1α via its binding to hypoxia-responsive elements located in the promoter region of the gene itself. The induction of FoxO1 proteins is indispensable for promoting the efficiency of antioxidant gene expression.

Here, we present a comprehensive analysis of the transcriptional response to HIF-1α, revealing the repression of numerous nuclear-encoded mitochondrial genes through the regulation of FoxO1 function. We demonstrate that through this signalling arm, HIF-1α reduces cellular ROS production, independent of MnSOD, catalase and Sesn3 activation. Regulation of mitochondrial structure and function could be an important role for HIF-1α factors in regulating ROS production, and these processes can affect cellular adaptation to hypoxia. Through in vitro drug experiments, we found that 2ME combined with As_2_O_3_ can inhibit MG63 cell proliferation and migration while promoting MG63 cell apoptosis and intracellular ROS accumulation. To further examine the effect of 2ME + As_2_O_3_, a xenograft murine model of OS in BALB/c nude mice was used to test its efficacy. In an in vivo drug-sensitivity test, the combination of 2ME and As_2_O_3_ achieved anti-tumour effects without obvious adverse reactions.

## 2. Materials and Methods

### 2.1. Microarray Data

We retrieved microarray data for normal tissues (GSM402687, GSM402688, GSM402689 and GSM402690) and human osteosarcoma tissues (GSM402693, GSM402703, GSM402731 and GSM402747) from the National Center for Biotechnology Information’s Gene Expression Omnibus (NCBI GEO) datasets for a total of eight samples.

**Ethical approval:** This study was approved by the Ethics Committee of Fudan University Shanghai Cancer Center.

### 2.2. Osteosarcoma Specimens

In all, 29 paired osteosarcoma specimens and adjacent normal bone tissues, which were confirmed as primary malignant bone cancer by trained pathologists, were collected from the Department of Musculoskeletal Oncology of the Fudan University Cancer Hospital (Shanghai, China) in 2017–2018. One of these samples was immediately snap-frozen in liquid nitrogen. The other tissues were formalin-fixed and paraffin-embedded.

### 2.3. Immunohistochemistry

Paraffin-embedded blocks were cut into 4 μm thick sections and dewaxed and hydrated. Then, the slices were immersed in distilled water containing 3% hydrogen peroxidase twice to reduce endogenous oxidase activity. Afterwards, the tissue sections were incubated with primary antibodies for 2 h at room temperature, and a secondary antibody was subsequently applied to the cells at room temperature for 40 min. The staining degree was determined by diaminobenzidine (DAB) chromogen (BioRad, Inc., Hercules, CA, USA). Subsequently, the tissues were dehydrated and sealed with gum. Five random fields of view (100×) were captured with a camera and a microscope (Olympus, Tokyo, Japan).

### 2.4. Cell Lines and Culture Conditions

Two human OS cell lines (U2OS and MG63) were purchased from the American Type Culture Collection (ATCC) and cultured in Dulbecco’s modified Eagle’s medium (DMEM) supplemented with 10% foetal bovine serum (FBS; Thermo Fisher Scientific, Waltham, MA, USA), 100 U/mL penicillin and 100 mg/mL streptomycin (Thermo Fisher Scientific). Regular osteoblast cells (hFOB1.19), used as a control, were acquired from the Chinese Cell Bank of the Chinese Academy of Sciences (Shanghai, China) and cultured in Ham’s F12/DMEM supplemented with 10% FBS, 100 U/mL penicillin and 100 mg/mL streptomycin. The cultures were preserved at 37 °C in a humidified CO_2_ (5%) atmosphere.

### 2.5. Quantitative Real-Time Polymerase Chain Reaction (qRT-PCR)

Complete RNA was extracted from cells and tissues with Trizol reagent (purchased from TAKARA, Tokyo, Japan) according to the protocol. All mRNA was subjected to quantitative polymerase reaction and reverse transcription according to the protocols for the PrimeScript^®^ RT Master Mix Perfect Real-Time Kit (TAKARA Bio Inc., Kusatsu, Japan) and SYBR Green Master Mix (Applied Biosystems, Foster City, CA, USA). qPCR was performed on an Applied Biosystems 7900HT Real-Time System (Foster, CA, USA).

### 2.6. Western Blot Analysis

Collected cells were lysed with RIPA protein extraction reagent (Beyotime, Beijing, China) containing a protease inhibitor cocktail (Roche, Pleasanton, CA, USA). The lysates were then loaded onto sodium dodecyl sulfate (SDS)-polyacrylamide gel electrophoresis (PAGE) gels for separation, transferred to polyvinylidene fluoride (PVDF) membranes and blocked in 5% milk prior to incubation with the indicated primary and secondary antibodies. Autoradiograms were quantified through densitometry, and GAPDH was used as a control. The antibodies against HIF-1α, FoxO1, MnSOD, catalase and Sesn3 were purchased from Cell Signaling Technology (Danvers, MA, USA).

### 2.7. Plasmid Construction and Cell Transfection

U2OS and MG63 cell lines were transiently transfected with siRNAs after being cultured in six-well plates overnight. A mixed negative control, a plasmid overexpressing FoxO1 and a blank vector were used as well with Lipofectamine 2000 transfection reagent (Thermo Fisher Scientific) and FuGENE^®^ HD transfection reagent (Roche, Wetzlar, Germany) according to the manufacturers’ instructions, respectively. The cells were then collected to observe the knockout or overexpression efficiency via qRT-PCR for 48 h after transfection. Two distinct siRNAs against FoxO1 were created and synthesized by GenePharma (Shanghai, China). The objective si-FoxO1 sequences and synthetic FoxO1 sequence (3099 bp) are described in previous research (15). The siRNA sequence targeting FoxO1 (si-FoxO1) was 5′-GCTCAACGAGTGCTTCATCAAGCTACCCA-3′.

### 2.8. Cell Proliferation Assay

Cell viability was determined with a CCK-8 assay. First, 1 × 10^3^ cells were seeded in quadruplicate for each group in a 96-well plate. The cells were incubated with 10% CCK-8 reagent (Dojindo Laboratories, Kimamoto, Japan) diluted in regular culture medium at 37 °C until optical colour conversion occurred. Proliferation rates were measured at 24, 48 and 72 h after transfection. The absorbance of each well was determined with a microplate reader (PerkinElmer, Waltham, MA, USA) set at 450 nm.

### 2.9. Xenograft Transplantation

Four- to six-year-old female nude mice were purchased from Vital River Laboratory Animal Technology (Beijing, China). All animals were housed in individual ventilated cages, provided sterilized water and food at libitum and handled under specific pathogen-free conditions in the Institute’s animal care facilities, which meet international standards. The mice were checked for their health status, animal welfare supervision was provided and experimental protocols and procedures were reviewed by a certified veterinarian. All animal experiments were carried out in accordance with the Chinese governing law on the use of medical laboratory animal (authorization no. 551998, 2013, by the Ministry of Health).

In total, 32 female mice were divided randomly into four groups. The average weight of the animals was 12 grams. The administered drugs were dissolved in dimethyl sulfoxide (DMSO). For the control group, the animals were injected with only DMSO intraabdominally. The other three groups were injected with a 2-ME/DMSO solution, an As_2_O_3_/DMSO solution and a 2-ME + As_2_O_3_/DMSO solution (2-ME at 5 mg/kg and As_2_O_3_ at 5 mg/kg). 2-ME was administered once every two weeks, and As_2_O_3_ was administered for five subsequent days and then every two days. The length of the treatment course was three weeks. Food and water were supplied ad libitum after treatment. The health status was monitored by a specific veterinarian. All animals were euthanized and sacrificed by cervical dislocation after treatment. After two courses of treatment, the mice were euthanized, and the tumours were removed and weighed.

### 2.10. Statistical Analysis

All statistical analyses were executed with SPSS 22.0 software (IBM Corporation, Armonk, NY, USA) and GraphPad Prism 5.0 (GraphPad Inc., La Jolla, CA, USA). Differences between groups were analysed utilizing Student’s *t*-test or one-way analysis of variance (ANOVA). Recurrence-free survival and total survival were determined by Kaplan–Meier survival analysis and compared via log-rank test. For this study, *p*-values <0.05 were considered statistically significant. Kyoto Encyclopedia of Genes and Genomes (KEGG) enrichment analysis was performed via the Database for Annotation, Visualization and Integrated Discovery (DAVID) program.

## 3. Results

### 3.1. HIF-1α and FoxO1 Expression was Increased in Human Bone Cancer Tissues

Differences and correlations were analysed for HIF-1α and ROS metabolic signalling pathway-related gene expression in normal tissues (GSM402687, GSM402688, GSM402689 and GSM402690) and osteosarcoma tissues (GSM402693, GSM402703, GSM402731 and GSM402747) from NCBI GEO datasets through R Package Limma and Affy. In all, 17 genes were discovered to be significantly differentially expressed in these datasets (Figure 1A). Among them, HIF expression was positively correlated with FoxO1, ACYP1, PPIH, PPIE and SESN1 and negatively correlated with CYP-related genes (Figure 1B). To further investigate their expression levels in human bone cancer, HIF-1α and FoxO1 expression was investigated in 29 paired OS and normal adjacent tissues using qRT-PCR. As depicted in Figure 1A, the 2^ΔΔCt^ values of HIF-1α and FoxO1 were significantly increased in bone cancer tissues relative to normal adjacent tissues (*p* < 0.05) (Figure 2A). This finding was consistent with the immunocytochemistry analysis. The immunocytochemistry assay results showed that HIF-1α and FoxO1 protein expression was significantly increased in bone cancer tissues compared with normal adjacent tissues (Figure 2B). Interestingly, HIF-1α interacted with FoxO1 through protein–protein association networks according to a STRING analysis (Appendix A).

### 3.2. ROS Production in OS Cells Was Related to HIF-1α Expression

Pyruvate dehydrogenase (PDH) activity is inhibited through HIF-1α by upregulation of the target molecule, pyruvate dehydrogenase kinase (PDK-1). HIF-1α blocks pyruvate in the tricarboxylic acid cycle, thereby inhibiting mitochondrial oxidative phosphorylation. Because mitochondrial respiration is the main source of ROS, we hypothesized that HIF-1α could reduce ROS production. Intracellular ROS levels were significantly lower in MG63 cells treated with HIF-1α siRNA than in control cells (Figure 3A,B). In addition, PDK-1 protein levels in MG63 cells transfected with HIF-1α-OE vector or HIF-1α siRNA and the corresponding empty vector transfected MG63 cells were determined via Western blotting. The results demonstrated that HIF-1α overexpression could promote the protein levels of PDK-1 (Figure 3C). qRT-PCR assay results further demonstrated that HIF-1α siRNA significantly reduced the mRNA expression of PDK-1 (Figure 3D).

### 3.3. HIF-1α Targeted FoxO1 Directly to Promote Its Expression in OS Cells

To further explore the role of HIF-1α in FoxO1-induced ROS changes, the effects of HIF-1α on FoxO1 expression were examined. Two hypoxia-responsive elements within the promoter region of FoxO1 were identified, indicating that FoxO1 was a potential HIF-1α target. The target sites in the *Homo sapiens* FoxO1 5′-UTR are shown in Figure 4A. To determine whether FoxO1 was a direct target of HIF-1α, reporter vectors containing the promoter region were constructed (ATG upstream 2000 bp); only one hypoxia-responsive element was retained, and full-length promoter deletions of FoxO1 were used (Figure 4A). Compared to the intact promoter region, full-length promoter deletion reduced luciferase plasmid activity by up to 60% (Figure 4A) and retaining only one hypoxia-responsive element reduced luciferase plasmid activity by up to 30% (Figure 4B). Furthermore, the relative DNA levels (Figure 4C) and ChIP-PCR (Figure 4D) results demonstrated that HIF-1α promoted FoxO1 expression by directly targeting FoxO1.

### 3.4. HIF-1α Expression Was Positively Correlated with FoxO1 and The Antioxidant Proteins MnSOD, catalase and Sesn3

The antioxidant proteins MnSOD, catalase and Sesn3 have been reported as primary downstream messengers of Akt/FoxO1 signalling [16]. To elucidate whether HIF-1α could regulate MnSOD, catalase and Sesn3 expression via FoxO1, their protein levels in MG63 cells transfected with HIF-1α-OE vector or HIF-1α siRNA and the corresponding NC cells were assessed via Western blotting. The results demonstrated that HIF-1α overexpression could increase the protein levels of FoxO1, MnSOD, catalase and Sesn3 (Figure 5A,B). Furthermore, HIF-1α overexpression increased the expression levels of FoxO1, MnSOD, catalase and Sesn3, but HIF-1α silencing decreased these expression levels in MG63 cells (Figure 5A,B). The qRT-PCR assay results further demonstrated that HIF-1α siRNA significantly reduced the mRNA expression of FoxO1, MnSOD, catalase and Sesn3 (Figure 5C). Next, HIF-1α-silenced MG63 cells were transfected with FoxO1-OE and the corresponding NC, and the FoxO1, MnSOD, catalase and Sesn3 protein levels were rescued, along with FoxO1 overexpression, in HIF-1α-silenced MG63 cells (Appendix A). Moreover, the cell migratory (Appendix A) and proliferation (Appendix A) capabilities were partially rescued along with FoxO1 overexpression in HIF-1α-silenced MG63 cells. As FoxO1 is a direct target gene of HIF-1α, all data demonstrated that MnSOD, catalase and Sesn3 were downstream effectors of HIF-1α that were at least partly induced by FoxO1 targeting.

### 3.5. The ROS Metabolism Regulation Pathway Is Involved in HIF-1α-Silenced OS Cells

To determine the mechanism of the relationship between HIF-1α and the ROS metabolism regulation pathway, gene expression was analysed in HIF-1α-silenced OS cells using a gene chip microarray. A heat map depicting gene expression in OS cells, as well as HIF-1α-silenced OS cells, indicated that the transcripts of five genes (*Cyp2c38*, *Cyp2c38*, *Cyp2c38*, *Ptgs2* and *Alox12*) participating in ROS metabolism were significantly increased in HIF-1α-silenced OS cells (Figure 6A), which was confirmed by qRT-PCR (Figure 6C). Furthermore, nine genes, including *Cat*, *Sod1*, *Sod2*, *Catalase*, *Nox4*, *Sesn3*, *FoxO1*, *Prdx1* and *Gpx1*, were significantly decreased in HIF-1α-silenced OS cells, which was also confirmed by qRT-PCR (Figure 6D). We further used Kyoto Encyclopedia of Genes and Genomes (KEGG) pathway analyses to determine the roles and corresponding molecular functions of the regulated genes in the pathways. The glycolysis pathway was enriched in the KEGG pathway analysis of upregulated genes (Figure 6B). These findings suggest that the ROS metabolism regulation pathway is closely involved in HIF-1α signalling in OS cells, and this hypothesis is consistent with previous reports on the important role of HIF-1α in OS [17].

### 3.6. ME + As_2_O_3_ Treatment Attenuates Tumour Growth

To evaluate the effect of epigenetic treatment on tumour growth, MG63 bone cancer cells were employed to create a tumour model. In these cells, ROS were induced by 2ME + As_2_O_3_; then, they were injected subcutaneously into nude mice. Beginning on day 7 after implantation, the tumour lengths and widths were evaluated every two days to acquire four measurements. Considerably slowed growth in the 2ME + As_2_O_3_ treatment group compared with that in the control group was revealed by the tumour growth curve (Figure 7A). The tumours were subsequently dissected, and their weights and accurate sizes were determined. Compared to those in the control group, the mass and average volume of the tumours were considerably lower in the 2ME + As_2_O_3_ treatment group (Figure 7B). Protein and total RNA were then extracted from each tumour and used to assess the expression levels of HIF-1α, FoxO1, Sesn3, MnSOD and catalase. After 13 days of xenograft growth in vivo, significant decreases in HIF-1α, FoxO1, Sesn3, MnSOD and catalase expression were exhibited by the tumours from the 2ME + As_2_O_3_ treatment group compared with the tumours from the control group (Figure 7D,E). During the treatment, no adverse effects were reported by the veterinarian who was responsible for monitoring the health of the animals. In summary, these outcomes imply that prior 2ME + As_2_O_3_ treatment attenuates tumour growth efficiently and may suppress OS growth by negatively regulating FoxO1 expression and HIF-1α.

### 3.7. ME + As_2_O_3_ Induce Bone Cancer Cell Death and Inhibit Migration

To determine whether 2ME + As_2_O_3_ can recapitulate the effects of epigenetic treatment, cell migration (Transwell) and proliferation (CCK8) assays were performed on MG63 cells. A Transwell assay revealed that the migratory capabilities of MG63 cells were greatly decreased when they were treated with 2ME + As_2_O_3_ (Figure 8A,B). Furthermore, a flow cytometry assay was performed to further detect the effect of 2ME + As_2_O_3_ treatment on bone cancer cell apoptosis. The results demonstrated that the group treated with 2ME + As_2_O_3_ had a significantly higher proportion of late/early apoptotic cells than the control group in MG63 cells (Figure 8C,D). CCK-8 assays revealed that cell proliferation was significantly inhibited following 2ME + As_2_O_3_ treatment in MG63 cells (Figure 8E). Taken together, these data indicate that 2ME + As_2_O_3_ exert proliferative and migratory effects on bone cancer cells and promote their apoptosis.

## 4. Discussion

OS cell apoptosis induced through hypoxia injury is a significant cellular biological event in this disease [23]. Until now, the underlying mechanism of hypoxia injury mediated by OS cells has not been completely understood. In the present research, we reported that FoxO1 and HIF-1α are primary players in OS cell hypoxic injury. Our data revealed that HIF-1α mediated FoxO1 expression under hypoxic tension. Moreover, activating the transcription factor HIF-1α induced ROS accumulation and simultaneously interrupted the antioxidant proteins MnSOD, catalase and Sesn3, which are crucial regulators of apoptosis in OS cells (Figure 8F).

FoxO factors are vital players in the regulation of target genes that are involved in apoptosis, differentiation and cell cycle [24,25]. Moreover, FoxO factors are also involved in the regulation of cellular reactions to oxidative tension [26]. In this study, we evaluated the expression of both Fox3a and FoxO1 in OS cells and unveiled a novel linkage between hypoxic injury and FoxOs. Our outcomes illustrated that hypoxia caused substantial increases in FoxO1 expression. Interestingly, FoxO1 expression continued to be fundamentally unchanged in hypoxic OS cells, meaning that FoxO1 might not be dispensable in hypoxic injury. FoxO transcription factors could include phosphorylation governed by Akt. FoxO1 can be stimulated by this phosphorylation to induce nuclear binding with the 14-3-3 protein, eventually giving rise to proteasomal degradation. In previous research, our outcomes indicated that hypoxia considerably decreases the expression of phospho-FoxO1 and phospho-Akt, suggesting that hypoxia impeded Akt activation in response to decreased FoxO1 phosphorylation. Furthermore, FoxO1’s intracellular localization was likewise influenced by hypoxia. FoxO1 was localized predominantly in the OS cellular cytoplasm under normal oxygen conditions, suggesting that FoxO1 was inactivated via degradation through the proteasomes. Nonetheless, hypoxia increased nuclear FoxO1 levels, which are related to HIF-1α activation, and this finding is consistent with the results of previous reports [27].

A verbatim protein–protein interaction between FoxO1 and HIF-1α has been revealed in a previous study, suggesting that FoxO1 may be a direct target of HIF-1α, although the protein that regulates FoxO1 synthesis continues to be unclear. Our outcomes disclosed that hypoxia increased the total protein expression and mRNA levels of FoxO1, which were abrogated through HIF-1α silencing. In contrast, HIF-1α silencing did not affect FoxO1 expression. Furthermore, the results were the same for phospho-FoxO1 expression in OS cells whether HIF-1α was silenced or not, indicating that HIF-1α did not require FoxO1 phosphorylation. Therefore, this information implied that HIF-1α increased FoxO1 protein synthesis via promoting transcription activity.

Physiological levels of ROS genesis within cells act as second messengers, which are presumed indispensable for normal cellular roles. Excessive ROS generation may prevail in a state of oxidative tension, which is related to pathophysiological alterations in many diseases [28,29]. The intracellular alterations when cells suffer ischaemia, including H^+^ and Ca^2+^ accumulation and the corresponding dysfunctions, affect the mitochondrial membrane potential, leading to ROS formation [30,31]. In addition to these points, stress–response pathway activation would be induced by ROS accumulation immediately, and apoptosis would be subsequently increased [32]. Simultaneously, many apoptosis stimuli, including treatment with tumour necrosis factor (TNF)-α [33] and lipopolysaccharide (LPS) [34] and growth factor withdrawal, can induce ROS generation through mitochondria [35]. In addition, antioxidants such as *N*-acetylcysteine, thioredoxin and MnSOD can mediate this effect and postpone apoptosis [36]. In summary, ROS have been viewed as one of the factors that can trigger and regulate apoptosis. In the present research, our outcomes elucidated that hypoxia facilitated ROS formation and the subsequent apoptosis increase in OS cells, which is in accordance with the hypothesis that oxidative tension plays a critical role in apoptosis in OS cells [37].

FoxO regulates genes that are proapoptotic, which are indispensable for inducing apoptosis. Cell death induced via ROS was also suppressed through a FoxO1 mutant [38]. Moreover, the FoxO family is inclusively modulated in response to oxidative tension, and ROS production is governed by FoxO1’s transcription activity [39]. These findings suggest that FoxO is a key regulator of ROS-induced apoptosis in mammalian cells. In the present study, we observed that FoxO1 knockdown via siRNA induced hypoxia-induced ROS accumulation in OS cells, indicating that hypoxia-induced ROS formation is required for FoxO1. However, these results are contrary to those of previous studies showing that FoxO1 regulates the detoxification of ROS [40] and the protective effects against cellular damage originating from oxidative stress [41]. One possible approach for explaining this difference is that the function of FoxO proteins has been reported to be highly dependent on context and may functionally vary among different cells and stimulation conditions. Therefore, the detailed relationship between FoxO1 and ROS stress requires further investigation.

Apoptosis acts as one of the primary pathways for programmed cell demise and can be activated via extracellular and intracellular factors. The components of the Bcl-2 family are this pathway’s decisive regulators of cellular demise, as they balance the ratio of anti- and proapoptotic proteins existing in mitochondria [42,43]. FoxO1 has been revealed to intervene in apoptosis via inhibiting objective genes, including Bim [44]. Moreover, a previous study illustrated that FoxO1 induces the proapoptotic proteins Noxa and Bim, which gives rise to cytochrome c release in neuroblastoma [45]. Bim also induces substantial ROS accumulation and mitochondrial respiration depletion [17]. In the present study, we found that hypoxia augmented apoptosis in OS cells, which was ablated by suppressing FoxO1 and HIF-1α. In addition, the proapoptotic proteins Bax and Bim EL increased, whereas the expression of Bcl-2 and antiapoptotic BclxL decreased under hypoxia, which is in accordance with a previous study’s findings. In addition, hypoxia induced Bcl-2 family protein disruption, which was likewise ablated through FoxO1 silencing. These data indicate that the Bcl-2 family intervention is an induction factor for hypoxia-induced apoptosis, and these family members are decisive FoxO3 downstream regulators.

In this study, we used an ex vivo cell model to dichotomize the regulatory loop of hypoxia injury in OS cells. Although the hypoxia model employed here is adequate for ruling out the confounding effects of neural and other humoural factors in vivo, the ischaemic conditions in animal models cannot be completely mimicked by this artificial model. Moreover, the relationship between ROS tension and FoxO1 was not extensively elucidated in this study. Further studies are necessary to illustrate the pathological mechanism in hypoxic OS cells.

2ME is a metabolite of oestradiol, which has a significant effect on proliferating cells but has no significant effect on resting cells. Therefore, the adverse reactions are small and have received much attention in recent years. At present, 2ME has entered the clinical trial stage and has good application prospects. Its main antitumour mechanisms include promoting apoptosis, increasing intracellular ROS content, arresting the cell cycle and inhibiting the formation of tumour blood vessels. As_2_O_3_ is a widely used drug. It can induce tumour cell apoptosis, the mechanism of which is as follows: it increases intracellular ROS levels and regulates apoptosis-related genes such as *c-myc*, *BCL-2*, *p53* and *caspases*. In our study, 2ME and As_2_O_3_ were used as HIF-1α inhibitors and ROS inducers to inhibit the proliferation and migration of MG63 cells, to promote the apoptosis of MG63 cells and to induce more ROS accumulation in cells. Furthermore, in nude mice, the combination of the two compounds can inhibit tumour growth and has a significant antitumour effect. However, the limitation lies in that detailed dosage information for these two drugs requires further experiments, and the safety window for dosage has not been determined within our study.

In summary, our information describes the substantive role of FoxO1 in hypoxia-induced ROS formation, which is perpetuated through HIF-1α. Moreover, FoxO1 induces an imbalance in antioxidant proteins (MnSOD, catalase and Sesn3), which then leads to OS cell apoptosis. This double consequence of FoxO1 appears to be crucial in hypoxic injury regulation in OS cells. The combination of As_2_O_3_ and 2ME can suppress the migration and proliferation of MG63 cells and increase MG63 cell apoptosis, thus inhibiting neoplasm development in vivo. Together, these data prove that HIF-1α-induced FoxO1 activation plays an important role in hypoxia-induced ROS accumulation and apoptosis in OS cells.

## Figures and Tables

**Figure 1 cells-09-00087-f001:**
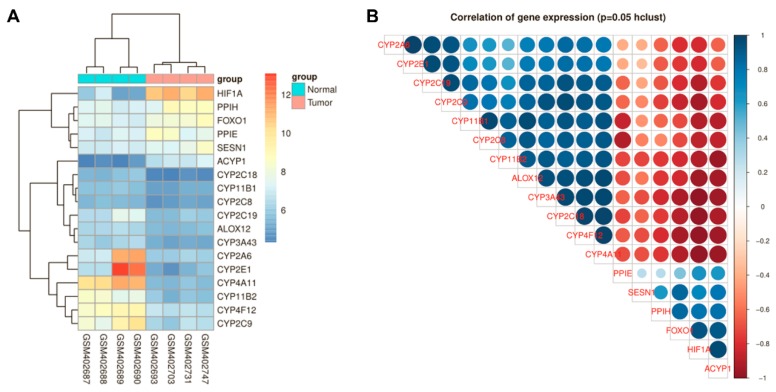
Correlation of gene expression for reactive oxygen species (ROS) metabolism regulation pathway in bone cancer tissues. (**A**) Differences in gene expression between normal tissues and bone cancer tissues. (**B**) Correlation of gene expression in bone cancer tissues. The correlation coefficient ranges from −1 (red colour) to +1 (blue colour). The red region represents absolute negative correlations. The blue region represents absolute positive correlations. Hclust, hierarchical clustering order. A value of 0.05 was chosen as the significance level.

**Figure 2 cells-09-00087-f002:**
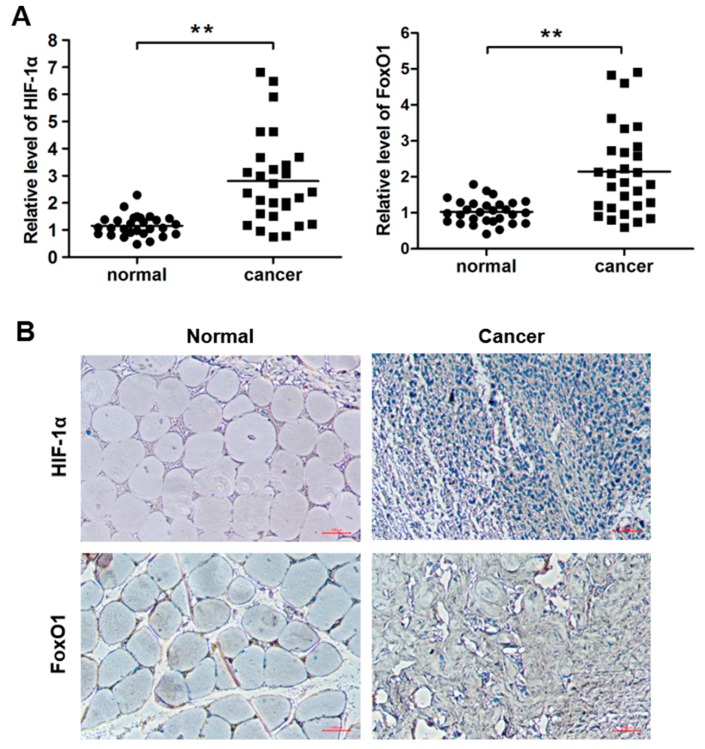
Hypoxia-inducible factor 1α (HIF-1α) and Forkhead box class O1 (FoxO1) expression is upregulated in bone cancer tissues. (**A**) HIF-1α and FoxO1 expression was analysed in bone cancer tissues and adjacent normal bone tissues via quantitative reverse transcription polymerase chain reaction (qRT-PCR). (**B**) Immunohistochemical analysis of HIF-1α and FoxO1 expression in adjacent normal bone tissues and bone cancer tissues. * *p* < 0.05; ** *p* < 0.01.

**Figure 3 cells-09-00087-f003:**
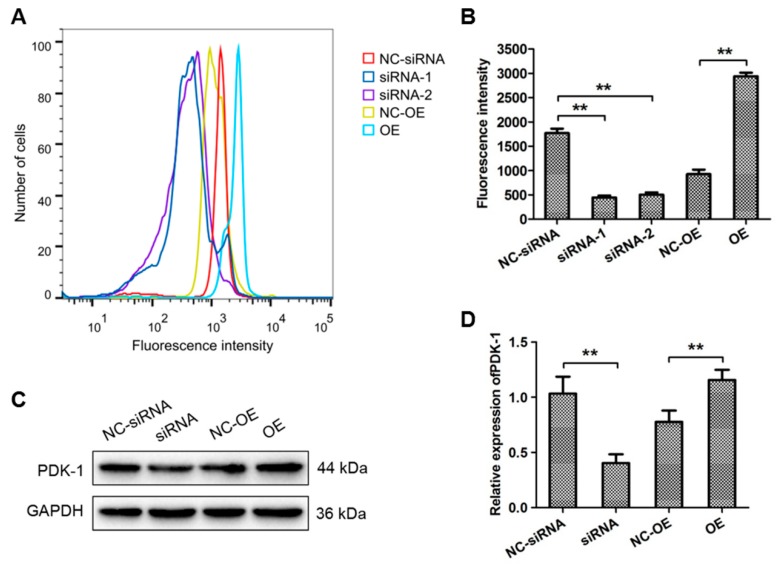
HIF-1α expression was positively correlated with pyruvate dehydrogenase kinase 1 (PDK-1) and inhibited ROS accumulation. (**A**,**B**) The intracellular ROS levels in MG63 cells treated with HIF-1α-OE and HIF-1α siRNA were analysed by flow cytometry. (**A**) Representative images; (**B**) quantitative analysis. (**C**) Western blot analyses were performed to evaluate PDK-1 protein expression in MG63 cells transfected with NC-siRNA, HIF-1α siRNA, NC-OE or HIF-1α-OE. (**D**) PDK-1 expression was then analysed via qRT-PCR. β-actin was used as a loading control. * *p <* 0.05, ** *p* < 0.01. NC, Negative Control; OE, HIF-1α-OE; siRNA, HIF-1α siRNA.

**Figure 4 cells-09-00087-f004:**
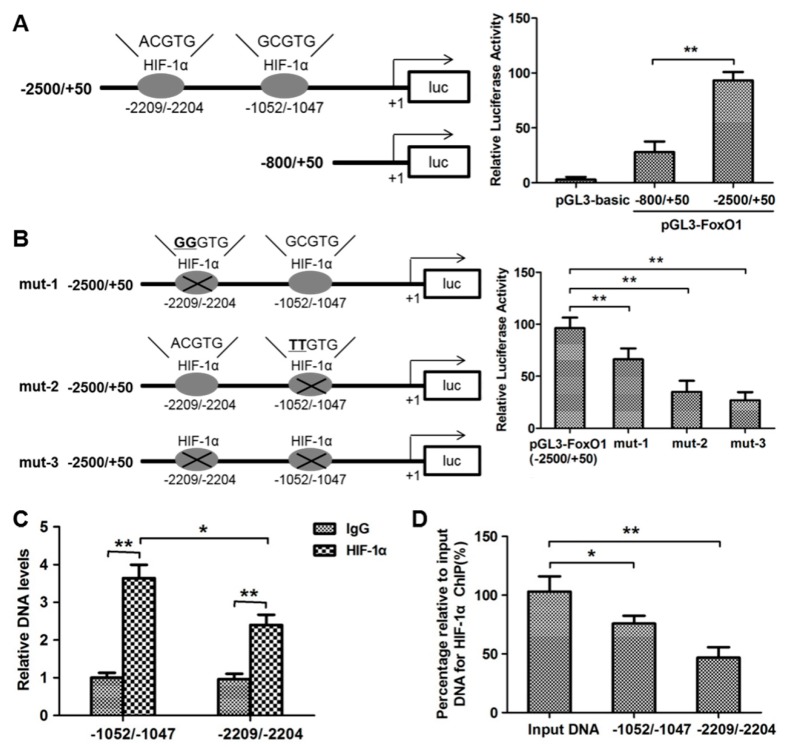
HIF-1α drives FoxO1 expression by binding directly to hypoxia-responsive elements within the promoter region of FoxO1. (**A**) A luciferase reporter plasmid encoding the promoter region (ATG upstream 2000 bp) and full-length promoter deletion of FoxO1 were transfected into MG63 cells. The luciferase activities were normalized to the β-galactosidase levels of the control. (**B**) Luciferase activities were detected for only one hypoxia-responsive element within the promoter region of FoxO1 in MG63 cells. (**C**) Relative DNA levels in the promoter region (ATG upstream 2000 bp) and full-length promoter deletion of FoxO1 were detected by PCR. (**D**) The percentage relative to the input DNA for HIF-1α was quantified by ChIP-PCR. * *p* < 0.05, ** *p* < 0.01.

**Figure 5 cells-09-00087-f005:**
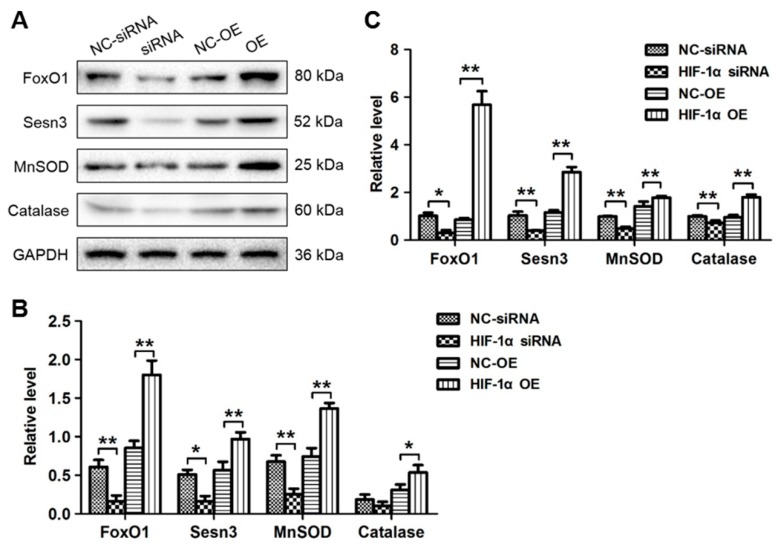
HIF-1α expression was positively correlated with FoxO1 and the antioxidant proteins MnSOD, catalase and Sesn3. (**A**) Western blot analyses were performed to evaluate FoxO1, MnSOD, catalase and Sesn3 protein expression in MG63 cells transfected with NC-siRNA, HIF-1α siRNA, NC-OE or HIF-1α-OE. (**B**) Quantitative analysis of the protein levels. (**C**) FoxO1, MnSOD, catalase and Sesn3 expression was then analysed via qRT-PCR. β-actin was used as a loading control. * *p* < 0.05, ** *p* < 0.01.

**Figure 6 cells-09-00087-f006:**
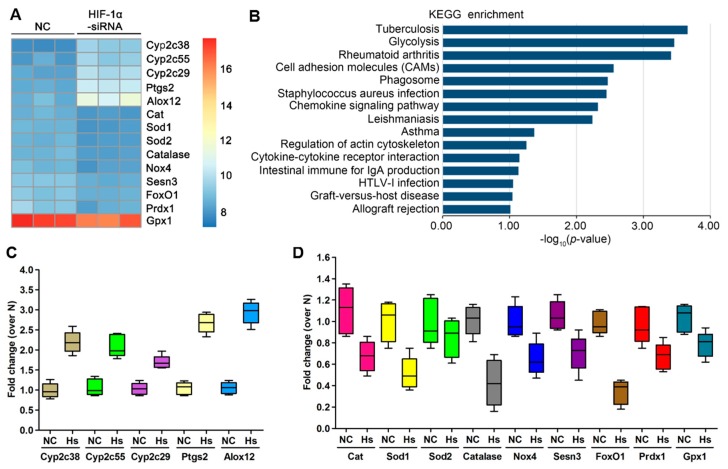
Gene expression profile of HIF-1α siRNA osteosarcoma (OS) cells. Gene expression profile of HIF-1α-silenced OS cells and control cells. (**A**) Heat map depicting the gene expression profiles of the ROS metabolism regulation pathway in HIF-1α siRNA OS cells. Red, high expression; yellow, intermediate expression; blue, low expression. (**B**) Top 15 canonical pathways enriched in the upregulated and downregulated genes, as determined by KEGG pathway analysis. (**C**) The expression levels of five upregulated genes were analysed by qRT-PCR. (**D**) The expression levels of nine downregulated genes were analysed by qRT-PCR.

**Figure 7 cells-09-00087-f007:**
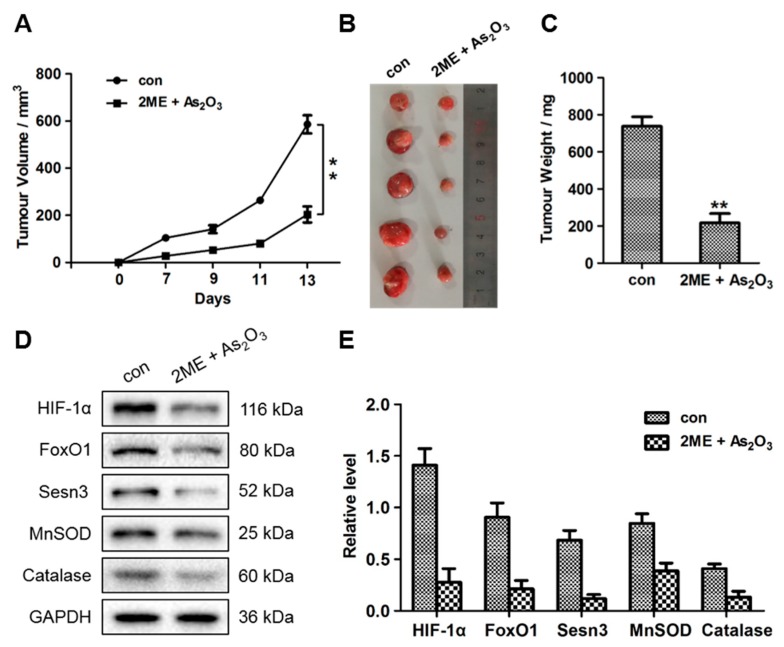
Effects of 2ME and As_2_O_3_ on OS xenograft growth in mice. (**A**) Representative images of the excised tumours of nude mice. (**B**,**C**) Quantitative analyses of the tumour weights (**B**) and volumes (**C**) (** *p* < 0.01). (**D**,**E**) Western blot analyses of HIF-1α, FoxO1, Sesn3, MnSOD and catalase protein levels in tumours from implanted mice. (**D**) Representative images; (**E**) quantitative analysis.

**Figure 8 cells-09-00087-f008:**
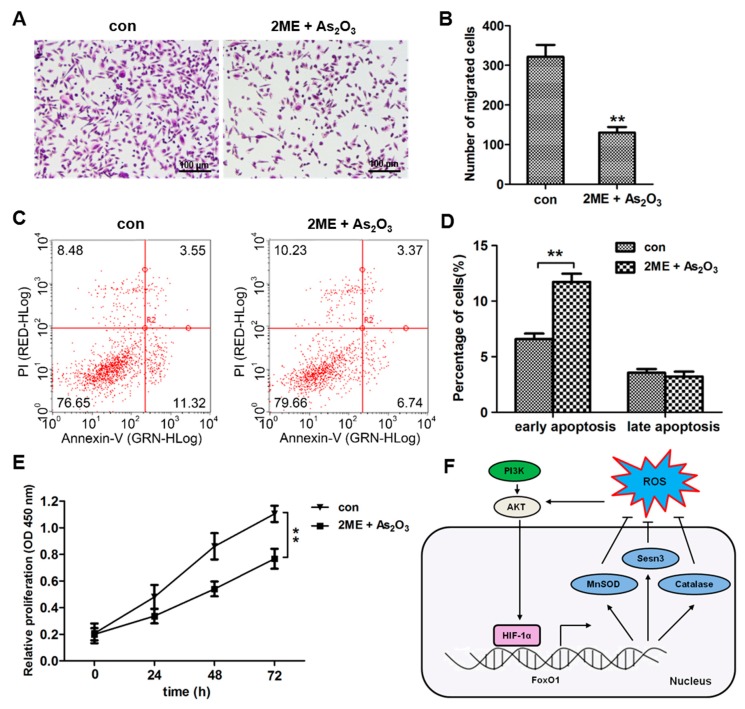
2ME and As_2_O_3_ inhibit the migration and proliferation of OS cells in utero. (**A**,**B**) Transwell analyses of the migrated MG63 cells and the cells treated with 2ME and As_2_O_3_. The cells were allowed to migrate for 12 h. (**C**,**D**) Apoptotic analyses of MG63 cells after treatment with 2ME and As_2_O_3_. The cells were stained with PI and analysed by flow cytometry. (**E**) A CCK-8 assay was performed to monitor the proliferation level of MG63 cells after treatment with 2ME and As_2_O_3_ at 0, 24, 48 and 72 h (* *p* < 0.05; ** *p* < 0.01). (**F**) Proposed scheme for the mechanism by which HIF-1α obstructs ROS accumulation in OS cells.

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
