# Peer review of "Metabolism of Reactive Oxygen Species in Osteosarcoma and Potential Treatment Applications"

_cells, 2019, doi:10.3390/cells9010087_

Round 1
Reviewer 1 Report
The authors have presented an interesting study on role HIF-1α and ROS osteosarcoma.
The results and the design of this study are very insightful, but I have the following concerns:
The English language of the paper can be improved. Although the results are focused on osteosarcoma but I think they should explain that role of ROS in cancer is very context-dependent, I suggest to mention the following works in the introduction or conclusion that in some cases of cancer such as lung cancer, reduction of ROS can lead to proliferation or cell death.https://insight.jci.org/articles/view/127647
https://febs.onlinelibrary.wiley.com/doi/abs/10.1002/1878-0261.12521
That can be more informative for the readers.
I also suggest the author to mention the role of mTOR signaling and autophagy in the introduction.
3. The authors need to explain the methods/programs they have used for pathway analysis (KEGG enrichment) and network analysis.
4."In vitro" and "In vivo" must be in Italic fonts in all sections.
Author Response
At first, we want to express our sincere gratitude towards the efforts from the reviewer.
We have made corresponding revision inside the manuscript to make the introduction more informative The topic of this article did not mention anything about mTOR signaling and autophagy。However, we alsp thanks for the comment We add a brief explaination for our KEGG enrichment analysis and network analysis We have made corresponding revision for the font.Reviewer 2 Report
In this manuscript, the authors realized a set of experiments that describes the metabolism of reactive oxygen species in osteosarcoma and potential treatment application.
Overall, this manuscript is well written, clearly organized and with good hypothesis. The experimental design is straight forward and utilizes appropriate methodologies.
This work is convincing although a few points should be considered:
-) The experiments were performed on a single cell line, the MG63 line. Some experiments, especially in vitro should be performed on one or two other lines of osteosarcoma.
-) Some controls should be shown as the efficacy of siRNAs against HIF-1a
-) In order to better understand the role of HIF-a, some experiments could be performed in hypoxia condition
Author Response
Thanks for your straight-forward comment, reviewer.
1.Our experiment only use the MG-62 cell as the model to illustrate our hypothesis. It is worthy to expand the idea to all the cell lines of osteosarcoma . However, we could compensate for it in the later study.
2.We have made corresponding revisions for it
3.The experiments in the hypoxia condition could be done in our later study, and we surely confirm that the role of HIF-1alpha could be discussed in future studies.
Round 2
Reviewer 2 Report
Accepted in present form